# Interlayer Difference of Bilayer-Stacked MoS_2_ Structure: Probing by Photoluminescence and Raman Spectroscopy

**DOI:** 10.3390/nano9050796

**Published:** 2019-05-24

**Authors:** Xiangzhe Zhang, Renyan Zhang, Xiaoming Zheng, Yi Zhang, Xueao Zhang, Chuyun Deng, Shiqiao Qin, Hang Yang

**Affiliations:** 1College of Advanced Interdisciplinary Research, National University of Defense Technology, Changsha 410073, China; xiangzhe.cheung@gmail.com; 2College of Arts and Science, National University of Defense Technology, Changsha 410073, China; 162201014@csu.edu.cn (X.Z.); zhangyi1983@zju.edu.cn (Y.Z.); yanghangnudt@163.com (H.Y.); 3College of Physical Science and Technology, Xiamen University, Xiamen 361000, China

**Keywords:** film–substrate interaction, photoluminescence, Raman spectroscopy, molybdenum disulfide, bilayer-stacked structure

## Abstract

This work reports the interlayer difference of exciton and phonon performance between the top and bottom layer of a bilayer-stacked two-dimensional materials structure (BSS). Through photoluminescence (PL) and Raman spectroscopy, we find that, compared to that of the bottom layer, the top layer of BSS demonstrates PL redshift, Raman $E_{2g}^{1}$ mode redshift, and lower PL intensity. Spatial inhomogeneity of PL and Raman are also observed in the BSS. Based on theoretical analysis, these exotic effects can be attributed to substrate-coupling-induced strain and doping. Our findings provide pertinent insight into film–substrate interaction, and are of great significance to researches on bilayer-stacked structures including twisted bilayer structure, Van der Waals hetero- and homo-structure.

## 1. Introduction

By stacking up two single-layer two-dimensional (2D) materials, bilayer Van der Waals (VdW) homo- and hetero-structures can be fabricated [1]. Owing to the existence of interlayer coupling, these bilayer-stacked structures usually exhibit distinct properties from their monolayer counterparts. For example, energy band gap evolution is found in bilayer VdW homo-structures compared to the corresponding monolayer, as previously reported in graphene and MoS_2_ [2,3]. Additionally, interlayer-coupling-induced p–n junction in VdW hetero-structure can lead to novel optoelectric effects [4,5]. Further, if stacking up two films with a misorientation angle, a brand-new tunable dimension is introduced to the bilayer-stacked two-dimensional materials structure (BSS), such BSS is referred to as twisted bilayer structure (tBLS). As a result, numerous exotic effects, induced by the twisted dimension and distinct from those in monolayer or bilayer without twisted angle, are expected. tBLS are tunable in their properties with variation in the twisted angle, thus have attracted intensive researches. For one thing, phonon in tBLS can be affected by interlayer coupling varying with angle, providing a simple but effective way to tune diverse properties such as, crystalline asymmetry [6], nonlinear optical effects [6,7], Raman scattering [8,9], and thermal conductivity [10,11,12,13,14,15]. For another, periodical interlayer Van der Waals potential can impact carrier performance of tBLS, which is first confirmed by the observation of Moiré pattern of twisted bilayer graphene (tBLG) under scanning tunneling microscope in 2005 [16]. Van Hove Singularity (VHS) [17] and angle-dependent electrical conductivity [18] are another two examples for this effect. Moreover, cutting-edge advances on twisted bilayer structure (tBLS) like, unconventional superconductivity in magic-angle tBLG [19] and mirror Dirac cone in incommensurate-angle tBLG [20], imply that there remains a lot that is yet to be explored.

However, all these findings about BSS focus only on the interlayer-coupling-induced effects, while ignoring the difference between the top and bottom layer of BSS. The BSS sample fabricated by transfer method can be divided into three different regions: stacked region where top and bottom layer overlap each other, bottom region (bottom layer excluding stacked region), and top region (top layer excluding stacked region). Although bottom and top regions are both supposed to be in direct contact with the substrate, there exists great difference between the top-substrate and the bottom-substrate coupling. Substrate coupling can affect 2D materials in many aspects. For example, on the one hand, substrate contact can employ strain on 2D materials, leading to phonon variations measured by Raman spectroscopy [21,22,23]. On the other hand, substrate can provide or deplete carriers depending on its doping type [24,25,26], thus tuning electrical and optical properties of materials deposited on it [27]. Furthermore, substrates with different permittivities and surface polar phonon modes demonstrate different scattering mechanisms limiting the electron mean free paths and mobility in 2D materials [28,29,30]. Band gap of semiconductors can also be tuned by dielectric environment permittivities [31]. Consequently, bottom and top region may demonstrate different phonon and exciton performance.

As one sort of transitional metal dichalcogenide (TMD) materials, MoS_2_ monolayer with a two-dimensional structure demonstrates intriguing effects in various aspects, including optical [32,33,34], electrical [35,36,37], and thermal properties [38,39,40]. Especially, due to its unique direct band gap [41], monolayer MoS_2_ is expected to have strong photoluminescence (PL) emission, which has been confirmed both experimentally and theoretically. Excited by 532 nm laser at ambient conditions, monolayer MoS_2_ is reported to have two prominent PL peaks at 625 nm (B peak) and 670 nm (A peak) [42,43,44]. These two peaks correspond to two direct excitonic transitions at the Brillouin zone K point, while the difference between them comes from the spin-orbital coupling caused by valence band energy splitting [42]. Also, two easily identified Raman peaks are observed in MoS_2_ monolayer [32,45], located near 390 cm^−1^ ($E_{2g}^{1}$, in-plane vibration mode) and 409 cm^−1^ (A_1g_, out-of-plane vibration mode), respectively.

Through photoluminescence and Raman spectroscopy, we found that, in bilayer-stacked MoS_2_ (BSM) samples fabricated by transfer, exciton and phonon performance in the top and bottom regions are remarkably different. Despite the fact that both top and bottom layers of BSM are transferred, compared to the bottom region, the top region demonstrates PL intensity reduction and peak redshift, implying less p-doping to top region from substrate. Meanwhile, redshift of in-plane Raman mode $E_{2g}^{1}$ is observed in top region, suggesting that vibration softens in top region. To exclude the film–substrate interaction, freestanding monolayer MoS_2_ samples are fabricated. It is found that, compared to the supported region, the suspended region of monolayer MoS_2_ demonstrates redshift in PL and Raman peaks, which are consistent with those in the top region of BSM, thus providing evidence for coupling difference between top-substrate and bottom-substrate.

Since the interlayer difference in BSS can complicate the experimental results, and is also affected by interlayer coupling, our findings are of great significance to distinguish between contributions from interlayer coupling and film–substrate interaction, which is of great significance to researches on interlayer-coupling-induced effects like optoelectric effects in VdW hetero-structure and twisted angle dependence in tBLS. Furthermore, our findings are universal and, apart from MoS_2_ bilayer-stacked structure on SiO_2_ substrate, we are sure this work can be informative to film–substrate interaction study on other 2D materials and substrates.

## 2. Materials and Methods 

### 2.1. Sample Preparation

For this study, a convenient fabrication process is employed to obtain BSM samples. First, all single-layer flakes are deposited on a SiO_2_/(001)Si substrate (SiO_2_ layer is 300 nm thick), with a size of approximately 50 μm by chemical vapor deposition (CVD). Then, we transferred two sheets of monolayer MoS_2_ to one substrate, by which method we can obtain tens of BSM samples with various angles in a single step. It should be noted that, in this work, both top and bottom layer of the BSM undergo transfer process to avoid preparation method induced difference. During the transfer process, any solvent that may cause doping in MoS_2_ was avoided. Universally used transfer methods like, PMMA-way (poly-methyl-methacrylate) [46] and PVA-way (poly-vinyl-alcohol) [47] introduce contamination or wrinkles to the surface of materials. Herein, a previously reported PLLA-way (poly-L-lactic-acid) [48] is chosen to ensure the transfer is residual-free and of high-uniformity. After transfer, the as-fabricated samples undergo ultraviolet treatment [49] and annealing [50] (in a tube furnace in Ar/H_2_ flow at 300 °C for 2 h) to remove residues and enhance interlayer coupling. 

As for free-standing monolayer MoS_2_ samples, they are fabricated by PLLA transfer onto SiO_2_/(001)Si substrate (SiO_2_ layer is 300 nm thick) with 300 nm-depth holes. These holes are of a radius 5 μm each, fabricated by reactive ion etching (RIE) method in SF_6_/CHF_3_ mixed gas flow (30 sccm).

### 2.2. Sample Characterization and Measurement

In this work, all bright field optical micrographs are taken by Nikon LV150 microscope, using 50× objective lens (Nikon, Tokyo, Japan). Dark-field optical micrographs are taken by ZEISS Axio Scope A1 microscope, using a 50× objective lens (Zeiss, Oberkochen, Germany). Atomic force microscopy (AFM) images are taken by NT-MDT Prima AFM system, using semi-contact scanning mode (NT-MDT, MoscowRussia). PL and Raman spectroscopy are measured by WITec Alpha300R confocal Raman system, using a 50× objective lens (WITec, Ulm, Germany). A 532 nm laser is used as the excitation source. For Raman measurements, the laser power is 1 mW, while for PL measurements laser power is 0.5 mW, sufficiently low to avoid heating effects. Optical gratings used for Raman and PL measurements are 1800 L/mm and 600 L/mm, respectively, providing respective spectral resolution smaller than 1 cm^−1^ and 1 nm.

## 3. Results

### 3.1. PL and Raman Difference between Layers of BSS

One as-fabricated BSM sample is shown in Figure 1. From the bright-field and dark-field optical micrographs in Figure 1a, we can see its surface is free of large-sized residual spots and wrinkles (of several micrometers size). To investigate its surface-height fluctuation in details, atomic force micrograph (AFM) is taken (Figure 1d). In Figure 1d, the sample’s surface seems bubble-free, uniform, and plane within each region (no sharp morphology fluctuations of several micrometers size). At the edge between the stacked and top region, where top layer falls from bottom layer to substrate, there seems no ramp but a vertical cliff.

As is plotted in Figure 1b, bottom, stacked, and top region of this BSM sample demonstrate easily distinguishable PL intensity. Bottom region demonstrates the strongest PL intensity, then followed by the top region and stacked region in turn. Meanwhile, Figure 1c shows spatial inhomogeneity within the top region. The area in the vicinity of the stacked region (V-area), outlined by magenta dashed line, exhibits lower intensity than rest of the top region. Moreover, there appears a general correlation between PL and Raman over the mapped area, i.e., this V-area can also be easily identified in Raman intensity map and Raman shift map, as is shown in Figure 1e,f respectively. In this V-area, compared to the rest of the top region, Raman mode $E_{2g}^{1}$ demonstrates redshift and intensity enhancement.

This difference in PL and Raman spectra between the top and bottom regions is also observed in other as-fabricated BSM samples, as is shown in Figure 2. For all samples in Figure 2b, the maximum $E_{2g}^{1}$-to-A_1g_ Raman shift difference among the top region and bottom region is below 19 cm^−1^, which is the signature of monolayer MoS_2_, indicating these samples are stacked by two individual monolayer MoS_2_. For each sample, compared to the bottom region, the top region exhibits PL intensity reduction and redshift (Figure 2a), and $E_{2g}^{1}$ Raman mode redshift (Figure 2b), indicating this interlayer difference in all samples shares a common origin.

### 3.2. Spatial Inhomogeneity in BSS

In addition, apart from the difference between the top and bottom layer, spatial inhomogeneity of PL emission and Raman scattering in the top region is prevalent among various BSM samples. Most importantly, it is found that, in many samples, area with lower PL intensity compared to the rest of the top region tends to emerge in the V-area. Another tBLS sample is shown in Figure 3, its PL intensity distribution on each region is consistent with the sample in Figure 1. It is noteworthy that an area (P2) outlined by pink dashed line in Figure 3c,e demonstrates identical PL intensity and $E_{2g}^{1}$ Raman shift with the V-area (P1) in this BSM sample. For more details, PL and Raman spectra on various regions are presented in Figure 3d,f respectively. Compared to the rest of the area of top region, PL emission of P1 and P2 demonstrates lower intensity and redshift, while Raman mode $E_{2g}^{1}$ also demonstrates redshift. Obviously, P1 and P2 are nearly same in PL and Raman spectra, implying identical exciton and phonon performance in these two regions. Moreover, though PL and Raman spectra in the V-area are remarkably different from rest of the top region (in Figure 1b–f and Figure 3c,e), their corresponding AFM micrographs (Figure 1d and Figure 3b) are spatially homogeneous. In contrast, the inhomogeneous area, with lower PL intensity, of bottom region (Figure 3c) matches exactly with the wrinkle and crack shown in the corresponding dark-field micrograph (Figure 3a inset). This implies that, the spatial inhomogeneity of PL and Raman spectra in top region is not due to abrupt variations in film morphology, like wrinkle, crack, and bubble.

### 3.3. PL and Raman of Freestanding MoS_2_

Interlayer difference and spatial inhomogeneity of PL and Raman spectroscopy in BSS possibly come from substrate-coupling difference between the layers, and among the top layer, respectively. To confirm this, we fabricated a freestanding sample (in Figure 4a) by transferring monolayer MoS_2_ to SiO_2_/Si substrate with holes of diameter 300 nm. The suspended (SUS) area of this sample totally excluded the film–substrate interaction. In Figure 4b–f, we can see that the suspended region demonstrates great intensity enhancement in PL and Raman spectroscopy, compared to that of the supported (SUP) region. This can be attributed to constructive interference effect in the top region [51]. Moreover, compared to the supported region, the suspended region demonstrates redshift in PL (Figure 4c,d) and Raman peaks (Figure 4g–i). This implies less p-doping and vibration mode softens in suspended region, which is consistent with the top region, P1 (V-area), and P2 in BSM. Therefore, similar to suspended region, we can assume that top region, V-area, and P2 might be less affected by substrate contact.

## 4. Discussion

A schematic illustration is shown in Figure 5a. According to our findings in Figure 4, less p-doping and vibration mode softening are observed in suspended region compared to its substrate-supported counterpart, which are also observed in top region, V-area, and P2 in BSM. Therefore, we speculate that, while bottom region and supported region are in strong coupling with the substrate, top region just like the suspended region is in intermediate or weak coupling with the substrate, thus leading to less carrier transfer and strain from substrate. In transferred-fabricated samples, this coupling mainly comes from Van der Waals bonding instead of chemical bonding [26,52]. In addition, as shown in Figure 5, there might be film morphology fluctuations in the top region, including ripple formed by strain and stair at the edge of the bottom region. Stair and ripple correspond to P1 (V-area) and P2 region in Figure 3 respectively. Though these film morphology fluctuations might be less than one nanometer (the order of monolayer MoS_2_ thickness), not sufficiently macroscopic to be detected by bright-field/dark-field optical microscope and atomic force microscopy, they can remarkably reduce film–substrate coupling. 

For one thing, PL emission of MoS_2_ is resulted from exciton (radiative wavelength: ~660 nm) and trion recombination (radiative wavelength: ~680 nm) [53]. According to related studies [24,25,26], the contribution ratio of exciton against trion determines the intensity and position of peak A. Since monolayer MoS_2_ is an n-type semiconductor, the silicon oxide depletes equilibrium electrons in regions of strong coupling with substrate [25], which stabilize the radiative recombination process of exciton (Γ_ex_) while suppressing the trion formation rate (Γ_f_) at the same time, as is shown in Figure 5b. PL emission, in regions of strong coupling with substrate (e.g., bottom region), is exciton-dominant, thus demonstrating intensity enhancement and blueshift. In contrast, areas of less coupling with substrate, such as top region, ripple, and stair, where exciton contribution is reduced, are supposed to demonstrate lower intensity and redshift of PL. 

For another, the in-plane Raman mode $E_{2g}^{1}$ corresponds to Mo and S atoms oscillating in the anti-phase parallel to the crystal plane, as shown in Figure 5c. As previously reported, $E_{2g}^{1}$ mode demonstrates redshift with uniform tensile uniaxial strain [21,22]. At the same time, it has been reported that film morphology fluctuations that lead to less coupling with substrate, like wrinkle and bubble, yield uniaxial tensile strain [54,55]. As a result, compared to the bottom region of strong substrate coupling, the top region of less substrate coupling is supposed to demonstrate $E_{2g}^{1}$ mode redshift caused by tensile uniaxial strain. Especially in ripple and stair regions, $E_{2g}^{1}$ is expected to demonstrate the strongest redshift. 

The discussions above provide a reliable explanation for our findings. Admittedly, interference effects can induce intensity change of PL and Raman. However, on the one hand, the minor height fluctuations in BSS film cannot result in remarkable interference variations. On the other hand, interference-induced intensity change would be broad band, which is not consistent with our experimental results. Therefore, we conclude that substrate-coupling-induced strain and doping to BSS play the dominant part in interlayer difference and spatial inhomogeneity of phonon and exciton performance.

## 5. Conclusions

In summary, we conducted a systematic investigation on PL and Raman spectroscopy of bilayer-stacked MoS_2_ fabricated by the transfer method. PL and Raman spectroscopy of freestanding monolayer MoS_2_ are also measured for comparison. Interlayer difference and spatial inhomogeneity of exciton and phonon performance are experimentally observed in as-fabricated BSS samples, which we attribute to film–substrate coupling-induced strain and doping. Additionally, our findings prove that, even surface fluctuations less than one-atom-layer thickness can be easily identified by Raman and PL spectroscopy. This work will be of great use to inform future researches on BSS including tBLS, VdW homostructure and heterostructure, and improve our understanding of substrate effects on optical and transport properties of 2D materials.

## Figures and Tables

**Figure 1 nanomaterials-09-00796-f001:**
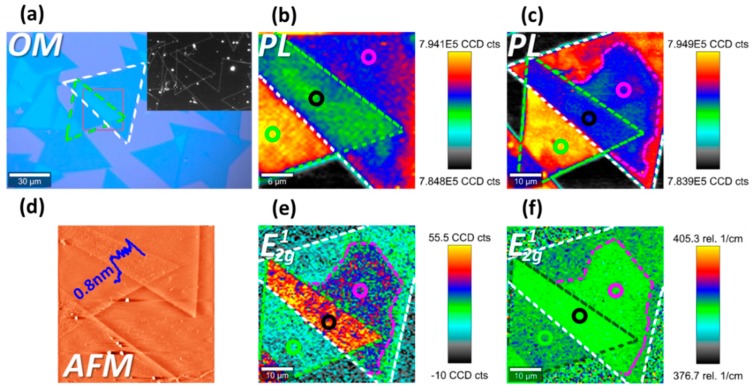
Interlayer difference of one twisted bilayer structure (tBLS) sample. Photoluminescence (PL) and Raman are all excited by 532 nm laser. (**a**) Optical micrograph. Inset corresponds to dark-field optical micrograph. The green and white dash lines outline the bottom and top layer respectively, while the red box outlines the scanning area in (**b**). (**b**,**c**) PL intensity map in high and low contrast respectively. (**d**) Atomic force microscopy (AFM) micrograph. (**e**) Raman intensity map of mode $E_{2g}^{1}$. (**f**) Raman shift map of mode $E_{2g}^{1}$. Green, black, and magenta circles in (**b**–**f**) point out bottom, stacked, and top region respectively.

**Figure 2 nanomaterials-09-00796-f002:**
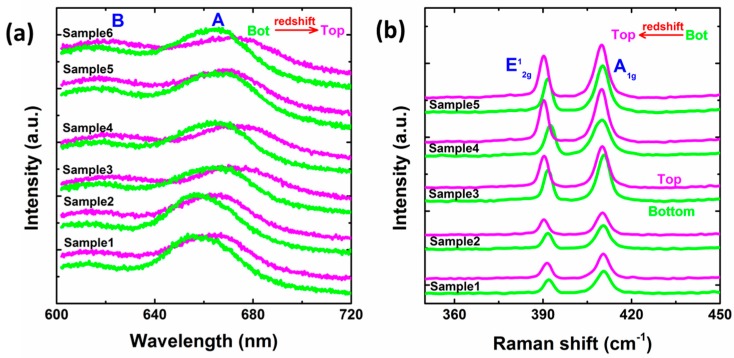
Photoluminescence and Raman spectra comparison between bottom (green) and top (magenta) layers. Bot and Top refer to bottom and top region, respectively. (**a**) PL spectra comparison. For clarity, spectra of one same tBLS sample are shifted vertically in small gap while spectra of different tBLS samples in large gap. Peak A and B are labeled. (**b**) Raman spectra comparison. Mode $E_{2g}^{1}$ and A_1g_ are labeled.

**Figure 3 nanomaterials-09-00796-f003:**
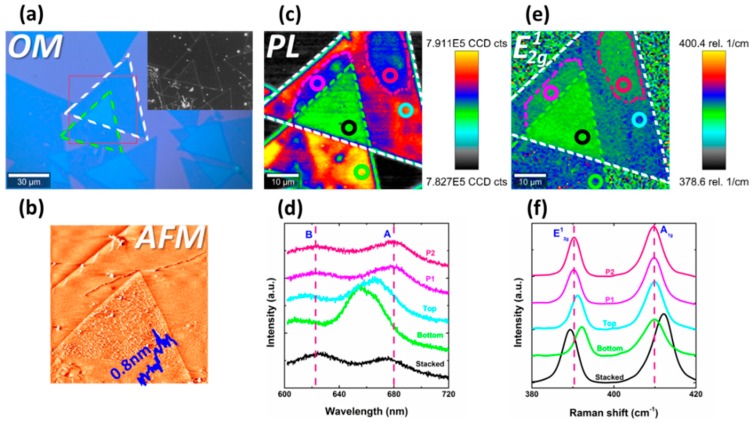
Photoluminescence and Raman inhomogeneity of tBLS. (**a**) Optical micrograph of one tBLS sample. Corresponding dark-field optical micrograph is shown in inset. Bottom and top layers are outlined by green and white dashed lines, respectively. Red box defines the scanning area of (**c**,**e**). (**b**) AFM micrograph of the same sample in (**a**). (**c**) PL intensity map. (**e**) Raman shift map of mode $E_{2g}^{1}$. (**d**,**f**) PL and Raman spectra of different regions. These regions are labeled by circles in corresponding colors in (**c**,**e**).

**Figure 4 nanomaterials-09-00796-f004:**
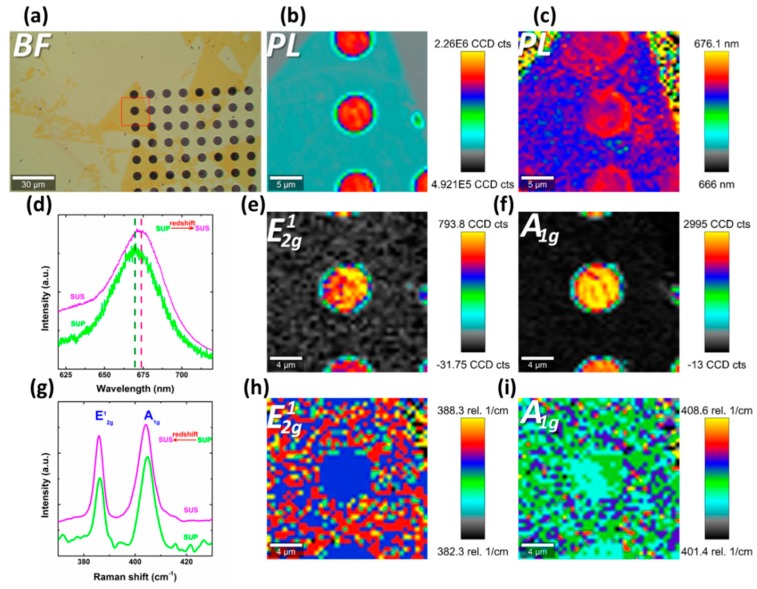
Free-standing monolayer MoS_2_. (**a**) Bright-field optical micrograph. Red box outlines the scanning area of the middle and right column. (**b**) PL intensity map. (**c**) PL shift map. PL and Raman spectra in (**d**,**g**) are normalized for clarity. (**d**) PL spectra of suspended and supported region. SUP and SUS refer to supported and suspended region, respectively. (**e**,**f**) Raman intensity map of $E_{2g}^{1}$ and A_1g_, respectively. (**g**) Raman spectra of suspended and supported region. (**h**,**i**) Raman shift map of $E_{2g}^{1}$ and A_1g_, respectively.

**Figure 5 nanomaterials-09-00796-f005:**
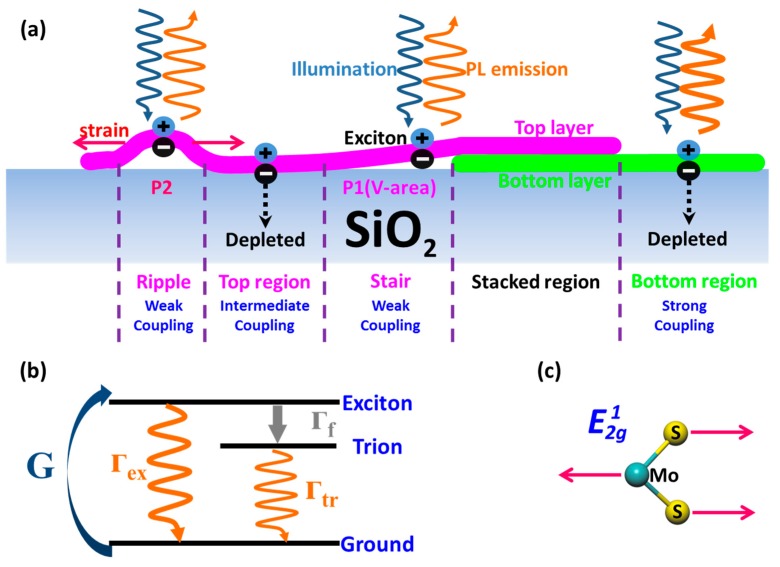
Schematic illustration. (**a**) Film–substrate coupling difference among bilayer-stacked two-dimensional materials structure (BSS) films. (**b**) Three-level energy diagram including exciton, trion, and ground. G represents the generation rate of exciton. Γ_ex_, Γ_f_, and Γ_tr_ represent exciton decay rate without trion formation rate, trion formation rate, and trion decay rate, respectively. (**c**) Schematic of in-plane Raman mode $E_{2g}^{1}$ for monolayer MoS_2_.

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
