# Peer review of "Interlayer Difference of Bilayer-Stacked MoS2 Structure: Probing by Photoluminescence and Raman Spectroscopy"

_nanomaterials, 2019, doi:10.3390/nano9050796_

Round 1

Reviewer 1 Report

The paper ‘Interlayer Difference of Bilayer Stacked Structure: Probing by Photoluminescence and Raman Spectroscopy’ investigates the influence of stacking in transferred bilayer MoS2 homo-structures on Raman and PL spectra. The work is important, particularly with the view on the ever-increasing body of work on stacked 2D materials. The work is of a specialized nature and as such the chosen journal is suitable. Unfortunately, there are a number off issues that need to be addressed to confirm the interpretation of the date. Since the work is generally important, I hope that the authors will address the following issues to ensure that the paper can be published. Major issues: 1. The authors state that certain sections of the overlapped 2D sheets are either the top and bottom sheet. The method with which this determination was carried out is not clear. Raman, PL and optical microscopy are not suited to assign which layer is on the top or bottom. The authors need to provide conclusive evidence of which layer is which. Even AFM is likely not suitable. One possible way could be to image the sample between the first and second transfer in order to label which sheet has been deposited first. 2. The authors state: “We can see its surface is free of residual, wrinkles or cracks. To investigate its surface height fluctuation in details, dark filed micrograph is taken (Figure 1d).” First of all, optical microscopy is not suited to determine if a 2D sheet is ‘free of residual, wrinkles or cracks’ and secondly, the optical dark field image can not detect fluctuations in height of 2D nanosheets (where fluctuations of interest are a few nm). The Authors should conduct AFM to confirm these statements or remove them. 3. Many of the assumptions in Fig 5 such as the existence of the ’stair’ region and the presence of wrinkles can be confirmed through AFM which would significantly strengthen paper. 4. The authors should provide optical images of the grown MoS2 flakes prior to the transfer to ensure that the sample only contained monolayers to start with. Sometimes bilayers grow during CVD synthesis and these bilayers could be transferred onto the sample giving the false impression that a stacked bilayer has been formed. Considering that a grown bilayer may very well have very different properties to a staked bilayer, this is important. 5. The authors did not provide any description regarding how the SiO2 substrate with holes was fabricated. Overall the Authors should conduct AFM of the analyzed area and provide optical images from after the first transfer and after the second transfer to confirm the identity of the top and bottom sheet.

Author Response

Please see detailed response in the submmited PDF file.

Response 1: We are very sorry for our negligence of detailed illustration of layer determination. During the fabrication, we investigate the profiles of MoS2 samples under optical microscope after first and second transfer to identify the layer. According to your comment, we have conducted AFM characterizations of the two BSS samples and add AFM images in Figure 1 and 3 in our revised manuscript, which can help identify top and bottom layer. 

Response 2: We are very appreciated for your comment. As you kindly point out, the optical bright field micrograph cannot effectively determine residual, wrinkles or cracks on 2D film. Actually, we mean in our manuscript that, optical bright field micrograph can just help detect remarkable residual spot, wrinkles and cracks (of several micrometers size). While the dark filed micrographs in our manuscript aim to confirm that interlayer difference comes not from sharp film morphology fluctuations (over several micrometers size). We are sorry for our unclear illustration in manuscript. We have modified this statement in our revised manuscript and add AFM characterizations to support it. As reviewer suggested, we add the AFM images in Figure 1 and Figure 3 of our revised manuscript. From AFM images (also shown below) we can see our samples are uniform in film morphology within each region (top, stacked and bottom region). The 0.8 nm height change can also help confirm that our BSS samples are stacked up by two individual monolayer films.

Response 3: Thanks for your constructive suggestion. We have added AFM characterizations in Figure 1 and 3 of our revised manuscript, from which we can see there exist no stair and wrinkle in film morphology, confirming our statement in manuscript that though the film morphology fluctuations are not large enough to be detected by bright-field/dark-field optical microscopy and AFM, they can effectively affect film-substrate coupling.

Response 4: This comment is informative to us. Monolayer MoS2 samples before transfer are shown in the following image. From the optical micrographs (Figure a and b) we can see, all MoS2 single crystalline flakes are same in optical contrast, implying that they are same in the number of layers. In Figure (c), the PL intensity map also confirm that the MoS2 single crystalline before transfer is spatial homogeneous in number of layers. Additionally, though Raman shift is spatially inhomogeneous among different region of our bilayer stacked samples, in Figure 2b and 3f, the maximum E1 2g-to-A1g Raman shift difference among top region and bottom region is below 19 cm-1, which is the signature of monolayer MoS2, indicating these samples are stacked by two individual monolayer MoS2. This discussion has been added to our revised manuscript.

Response 5: Thanks for your suggestions. The silicon oxide substrate with holes is fabricated by reactive ion etching method (RIE) in SF6/CHF3 mixed gas flow (30 sccm). The holes are at depth of 300 nm, radius of 5μm. We have added this description in our revised manuscript in section 2.1. Sample preparation.

Reviewer 2 Report

This paper reports a micro-Raman and micro-photoluminescence investigation of bilayer stacked MoS2 structures, obtained by sequential transfer of monolayer MoS2 flakes onto a SiO2 substrate. Noteworthy, different Raman and PL signals are obtained not only in the region where the 2D layers are overlapped, as a consequence of interlayer coupling. In fact, differences are also observed between the top and bottom layer regions over the substrate (SiO2), with a red shift of the Raman and PL spectra in the case of the top layer. This was ascribed to a lower coupling of the top layer with the SiO2 substrate with respect to the bottom one, which results in a reduced p-type doping and vibration mode softening. This explanation was supported by red-shift of Raman and PL spectra of free-standing monolayer MoS2 over holes in the SiO2 substrate.

The paper is well written and the results can be certainly of interest for the 2D materials community. I support publication in Nanomaterials, after the following points will be addressed.

1.     I suggest to perform morphological images of the bilayer stacked MoS2 by atomic force microscopy, to support the configuration of the layers indicated in the schematic of Fig.5(a).

2.     Although the reported results can be, in principle, generalized for any bilayer stacked structure, this investigation is focused on MoS2 bilayer stacks. Hence, it would be better to change the title into: “Interlayer Difference of Bilayer Stacked MoS2 Structure: Probing by Photoluminescence and Raman Spectroscopy

3.     As correctly stated by the authors in the paper’s Introduction, substrate coupling can affect 2D materials in many aspects. The effects are not only on strain and doping. As an example, different scattering mechanisms limiting the electron mean free paths and mobility in 2D materials come into play for substrates with different permittivities and surface polar phonon modes (see, as references, Nano Lett. 11, 4612–4618 (2011); Nat. Nanotechnol., 6, 147–150 (2011); Adv Mater. 28, 547-52 (2016)). Furthermore, in the case of 2D semiconductors, bandgap engineering can be achieved playing on the permittivity of the dielectric environment (see, e.g., Nature Electronics, 2, 54-55 (2019), and references therein).

For the sake of generality, these aspects should be mentioned in the Introduction.

Author Response

Response 1: We are appreciated for your constructive suggestion. We have added atomic force microscopy images of the selective samples in Figure 1 and 3 of our revised manuscript, from which we can evidently see no sharp film morphology fluctuations within each region of BSS sample, implying our conclusion that the interlayer difference cannot come from macroscopic surface height change.

Response 2: Thanks for your kind comment. Your suggestion title is obviously more appropriate for the main idea of our work. We have modified the title in our revised manuscript according to your comments.

Response 3: These suggested related studies are very informative to our work. We have cited and discussed these work in section 1. Introduction of our revised manuscript.

Please see detailed reposne in the submitted PDF file.

Round 2

Reviewer 2 Report

The authors addressed all the reviewers' questions and significantly improved the manuscript.

The paper can now be accepted for publication.